

# Intra-individual heteroplasmy in the *Gentiana tongolensis* plastid genome (Gentianaceae)

Shan-Shan Sun[1], Xiao-Jun Zhou[1], Zhi-Zhong Li[2,3], Hong-Yang Song[1], Zhi-Cheng Long[4] and Peng-Cheng Fu[1]

[1] College of Life Science, Luoyang Normal University, Luoyang, Henan, People's Republic of China
[2] Key Laboratory of Aquatic Botany and Watershed Ecology, Wuhan Botanical Garden, Chinese Academy of Sciences, Wuhan, Hubei, People's Republic of China
[3] University of Chinese Academy of Sciences, Beijing, People's Republic of China
[4] HostGene. Co. Ltd., Wuhan, Hubei, People's Republic of China

## ABSTRACT

Chloroplasts are typically inherited from the female parent and are haploid in most angiosperms, but rare intra-individual heteroplasmy in plastid genomes has been reported in plants. Here, we report an example of plastome heteroplasmy and its characteristics in *Gentiana tongolensis* (Gentianaceae). The plastid genome of *G. tongolensis* is 145,757 bp in size and is missing parts of *petD* gene when compared with other *Gentiana* species. A total of 112 single nucleotide polymorphisms (SNPs) and 31 indels with frequencies of more than 2% were detected in the plastid genome, and most were located in protein coding regions. Most sites with SNP frequencies of more than 10% were located in six genes in the LSC region. After verification via cloning and Sanger sequencing at three loci, heteroplasmy was identified in different individuals. The cause of heteroplasmy at the nucleotide level in plastome of *G. tongolensis* is unclear from the present data, although biparental plastid inheritance and transfer of plastid DNA seem to be most likely. This study implies that botanists should reconsider the heredity and evolution of chloroplasts and be cautious with using chloroplasts as genetic markers, especially in *Gentiana*.

## INTRODUCTION

Chloroplasts transform solar energy into chemical energy stored in the form of carbohydrates, supporting life on Earth. In angiosperms, plastomes are generally highly conserved with a quadripartite structure that contains a pair of inverted repeats (IRs) that separate the remaining regions into one small single-copy region (SSC) and one large single-copy region (LSC) (*Palmer, 1985*; *Jansen et al., 2005*). Plastomes range in size from 120 to 160 kb and contain conserved gene content and order, usually 110–130 genes in angiosperms (*Palmer, 1985*; *Ruhlman & Jansen, 2014*). Most flowering plants exhibit maternal plastid inheritance, and rarely paternal plastid inheritance (*Schumann & Hancock, 1989*), but some genera display biparental plastid inheritance to some degree (*Mogensen, 1996*). At the level of population or species, plastids exhibit sequence and

Corresponding authors
Xiao-Jun Zhou, lynubio@126.com
Peng-Cheng Fu, fupengc@lynu.edu.cn

structural polymorphisms (*Sun et al., 2018*), but heteroplasmy, the condition of cells having more than one organelle haplotype, has also been reported in limited taxa (*Fitter et al., 1996*; *García, Nicholson & Nickrent, 2004*; *Frey, Frey & Forcioli, 2005*; *McCauley et al., 2007*).

Chloroplast data are ubiquitous in plant systematics research due to its uniparental inheritance, haploid nature, highly conserved structure, and slower evolutionary rate of change compared to nuclear genomes (*Wolfe, Li & Sharp, 1987*; *Shaw et al., 2014*). Therefore, the plastome has been widely applied to reassess classifications, divergence dating, mutational hotspot identification, and genome evolution (*Nikiforova et al., 2013*; *Yao et al., 2019*). The National Center for Biotechnology Information (NCBI) contains more than 7,000 records of complete chloroplast genomes in plants as of April 24th, 2019 (title = chloroplast AND complete genome).

As the largest genus in family Gentianaceae, *Gentiana* contains 15 sections and 362 species (*Ho & Liu, 2001*). *Gentiana* is predominantly alpine around the world, and some taxa have already been cultured (*Rybczyński, Davey & Mikuła, 2015*) due to chemical and horticultural value (*Ho & Liu, 2001*). The mountain ranges surrounding the Qinghai-Tibetan Plateau (QTP), where roughly 250 species exist (*Ho & Pringle, 1995*), are the main diversity center of *Gentiana* (*Favre et al., 2016*). Presently, the plastomes of 12 *Gentiana* species have been sequenced (*Fu et al., 2016*; *Ni et al., 2016*; *Wang et al., 2017*; *Sun et al., 2018*), which belong to two sections, *Cruciata* Gaudin and *Kudoa* (Masamune) Satake & Toyokuni ex Toyokuni. In addition to these plastomes, a number of plastid fragments have been sequenced and used to reconstruct phylogenetic relationships in *Gentiana* and Gentianaceae (*Favre et al., 2016*). All previous studies tacitly accepted the maternal inheritance and haploid nature of plastomes in these species, and there are no reports regarding heteroplasmy in Gentianaceae.

As part of our ongoing study of *Gentiana* plastomes, some surprising heteroplasmy was uncovered in *G. tongolensis* Franchet, which is an annual herb belonging to section *Microsperma* T.N. Ho series *Suborbisepalae* Marquand (*Ho & Liu, 2001*). This species is endemic to China with its distribution limited to the southeast of the QTP (*Ho & Liu, 2001*). *Gentiana tongolensis* has been a poorly studied species within a large genus, except for its classification and taxonomy (*Favre et al., 2016*). Here, we report heteroplasmy in the plastome of *G. tongolensis* at the nucleotide level. This information will help botanists to reconsider the heredity and evolution of chloroplasts and to take caution with their use as genetic markers.

## MATERIALS AND METHODS

### Sample collection, genome sequencing and assembly

Five individuals of *G. tongolensis* were sampled at Daocheng, Sichuan Province in the QTP (28°52′E, 100°16′N). The species was identified by Dr. Peng-Cheng Fu, and its voucher specimens (No. Fu2016162) were deposited in the herbarium of the College of Life Science, Luoyang Normal University. Each sample was collected from a single plant and fast dried by silica gel. One sample (Fu2016162.05) was used for genome sequencing in this study. We performed total genomic DNA isolation, DNA fragmentation, and

sequencing library construction following the process described by *Fu et al. (2016)*. The extracted total genomic DNA was quantified by a Qubit3.0 fluorometer (Thermo Fisher Scientific, Waltham, MA, USA) and 1% agarose electrophoresis. The fragmented genomic DNA was sequenced using the Illumina HiSeq 4000 platform (Novogene, Tianjing, China) yielding 150-bp paired-end reads from a library of around 300-bp DNA fragments. Based on the genome size (*Mishiba et al., 2009*) and previous plastid genome studies (*Fu et al., 2016*; *Sun et al., 2018*) in *Gentiana*, we obtained approximately seven Gb of raw data that was deposited in the Genome Sequence Archive of the BIG Data Center (Accession No. CRA001588). The raw reads were processed in a relatively standard manner in Trimmomatic 0.38 (*Bolger, Lohse & Usadel, 2014*), beginning by trimming and removing bases with qualities less than 20 at both 5′ and 3′ ends, and reads with less than 36 bp and average base quality less than 15 in a 4-base window. The plastome was de novo assembled in NOVOPlasty 2.6.1 (*Dierckxsens, Mardulyn & Smits, 2016*). The plastome sequence of *G. tongolensis* was deposited in GenBank (Accession No. MK251985).

## Genome annotation and comparative analysis

The protein coding genes, rRNAs, and tRNAs in the plastome were predicted and annotated using Dual Organellar GenoMe Annotator (*Wyman, Jansen & Boore, 2004*) and GeSeq (*Tillich et al., 2017*). Comparative analysis was performed with three *Gentiana* species that represent the different groups reported (*Sun et al., 2018*). Plastomes were downloaded from NCBI (*G. straminea* NC_027441, *G. lawrencei* var. *farreri* KX096882, *G. stipitata* NC_037984), and genome comparisons were performed to identify the differences among taxa using Geneious Basic 5.6.4 (*Kearse et al., 2012*) and mVISTA (*Frazer et al., 2004*).

## Single nucleotide polymorphism calling and verification

Reads used for de novo assembly were aligned to the assembled plastome using BWA-MEM (*Li, 2013*) with default parameters. Single nucleotide polymorphisms (SNPs) or indels were then called using Pisces 5.2.11 (*Dunn et al., 2019*) with default parameters. Pisces is an Illumina variant caller for calling low frequency variants in tumors by producing a Q-score indicating the confidence that the variant is indeed present. SNPs or indels with variant frequency less than 2% were omitted to avoid sequencing errors, whose rate can approach 1–2% in some situations (*Schirmer et al., 2016*). For reducing the effect of low-quality bases, SNPs with a Q-score of 100 were chosen for downstream analysis.

For verifying the SNPs in the plastome, three regions (GTcpSNP-3, GTcpSNP-4 and GTcpSNP-5) located in the LSC with high SNP frequencies were chosen (Fig. 1). GTcpSNP-3, located at the IRa-LSC boundary and ranged from 145,647 to 664 in the plastome, includes parts of *rps19*, all of *trnH*, and parts of *psbA*. GTcpSNP-4, ranging from 23,015 to 23,631 in the plastome, is part of gene *rpoB*. GTcpSNP-5, ranging from 68,312 to 68,882 in the plastome, is part of gene *psbB*. Another two regions (GTcpSNP-1 and GTcpSNP-2) with no SNPs were also chosen as controls (Fig. 1). GTcpSNP-1 is located in the SSC region and ranges from 114,570 to 115,206 in the plastome. GTcpSNP-2 is located in the IR region and ranges from 129,395 to 129,965 in the plastome. Verification was

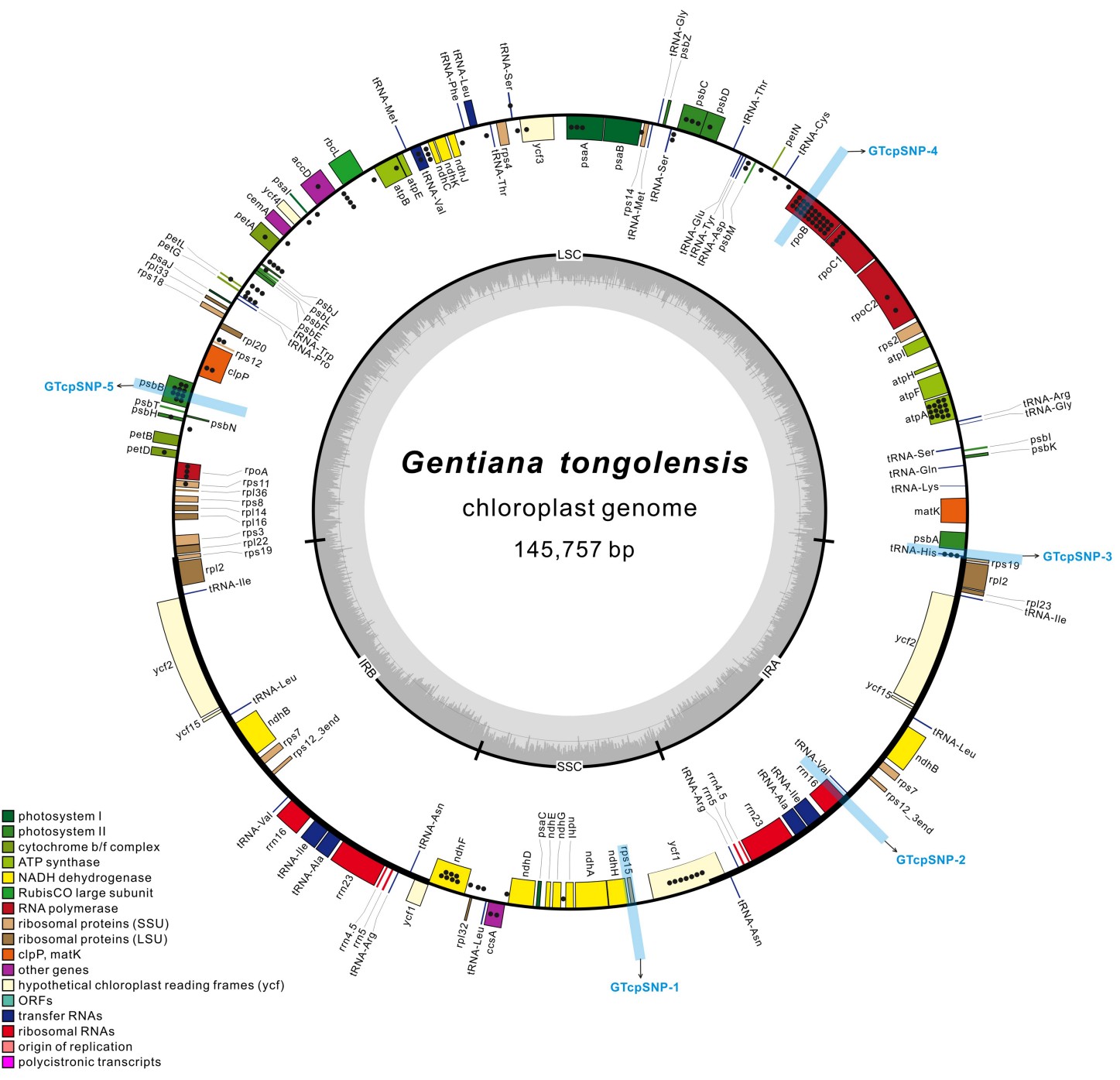

**Figure 1 The structural map of *Gentiana tongolensis* plastome.** Genes drawn inside the circle are transcribed clockwise, and those outside are transcribed counterclockwise. Genes belonging to different functional groups are shown in different colors. Dots represent SNPs or indels detected from Illumina data in a single individual of *G. tongolensis*. Detailed information about SNPs or indels are presented in Table S2. Five regions (from GTcpSNP-1 to GTcpSNP-5) used for verification are indicated by light blue boxes.

performed first in the same individual that was sequenced to obtain the Illumina data and then four more individuals in the same population. Since primers could not be designed for the loci for TaqMan, the method of transformed cloning was adopted.
**Table 1 Basic information of the plastome and its SNPs or indels in *Gentiana tongolensis*.**

|  | Length (bp) | No. | Percentage of polymorphic sites (%) |
|---|---|---|---|
| LSC | 78,289 | 121 | 84.62 |
| IR | 25,359 | 0 | 0 |
| SSC | 16,750 | 22 | 15.38 |
| Total | 145,757 | 143 | 100 |

Primers were designed using PRIMER version 5.0 (*Clarke & Gorley, 2001*), and their information is presented in Table S1. PCR was performed in 25 mL volumes containing 20 ng of template DNA, 1× PCR Buffer, 1.5 mM MgCl$_2$, 0.25 mM of each dNTP, 0.3 mM of each primer, and 1 unit of *Taq* DNA polymerase (Takara, Dalian, China). The PCR cycling profile included an initial step of 5 min at 95 °C, followed by 35 cycles of denaturation at 95 °C for 50 s, 50 s of annealing at 54 °C, and 30 s at 72 °C, and a final extension at 72 °C for 7 min. PCR products were purified by an eZNA DNA Gel Extraction Kit (Omega Bio-Tek, Guangzhou, China). The concentration of purified PCR products was measured by a NanoDrop 2000c Spectrophotometer (Thermo Scientific, Waltham, MA, USA), and then the PCR products were ligated into a pMD19-T vector (Takara, Dalian, China) and transformed into Trans5α Chemically Competent Cells (TransGen, Beijing, China). Positive clones were tested in a 20 μL PCR reaction volume containing 10–100 ng template DNA, 1× PCR Buffer, 1.5 mM MgCl$_2$, 0.2 mM of each dNTP, 0.2 mM of M13F/R, and 1 unit of *Taq* DNA polymerase (Takara, Dalian, China). PCR was performed under the following program: an initial step of 5 min at 95 °C followed by 20 cycles of 30 s at 95 °C, 1 min at 53 °C, and 30 s at 72 °C, followed by a final extension step at 72 °C for 7 min. The positive clones were sequenced using the Sanger method with M13 universal primer.

### Genetic analysis of verified loci

Sequences were aligned with Geneious Alignment and edited in Geneious Basic 5.6.4 (*Kearse et al., 2012*). Haplotypes were identified in DnaSP 5.1 (*Librado & Rozas, 2009*) and deposited in GenBank (MK251953–MK251984, MN602314–MN602324). Phylogenetic relationships among haplotypes were reconstructed in PhyML 3.0 (*Guindon & Gascuel, 2003*) with the GTR model estimated in jModelTest 2.1.7 (*Darriba et al., 2012*).

## RESULTS

### Quality of reads and features of the plastome

There were 218 million clean reads counting for 18.3 billion bases from high-throughput sequencing. The Q20 and Q30 were 99.99% and 96.69%, respectively. The size of the plastid genome was 145,757 bp in length (Table 1). Its circular-mapping assembly possessed the typical quadripartite structure (*Shinozaki et al., 1986*; Fig. 1) and comprised a pair of IR regions (IRa and IRb), one LSC region, and one SSC region, which were 25,359 bp, 78,289 bp, and 16,750 bp, respectively. A comparison of plastomes in

**Table 2 The 20 SNPs or indels with highest frequencies in the plastome of a single individual of *Gentiana tongolensis*.**

| Position | Reference base | SNP | Depth of coverage | Frequency of SNP | Location | Region |
|---|---|---|---|---|---|---|
| 14 | A | G | 2843 | 0.310 | *trnH* | LSC |
| 30 | A | G | 3265 | 0.297 | *trnH* | LSC |
| 54 | A | G | 3794 | 0.281 | *trnH* | LSC |
| 68516 | CTGTAAA | C | 4405 | 0.255 | *psbB* | LSC |
| 68943 | G | A | 3819 | 0.201 | *psbB* | LSC |
| 69026 | G | A | 3780 | 0.200 | *psbB* | LSC |
| 38948 | T | G | 3792 | 0.184 | *psaA* | LSC |
| 68569 | A | G | 3998 | 0.171 | *psbB* | LSC |
| 68436 | C | T | 3866 | 0.169 | *psbB* | LSC |
| 23192 | A | G | 3545 | 0.155 | *rpoB* | LSC |
| 68769 | GATTGAATT | G | 3672 | 0.153 | *psbB* | LSC |
| 22457 | A | G | 3518 | 0.149 | *rpoB* | LSC |
| 23411 | C | T | 3586 | 0.145 | *rpoB* | LSC |
| 9086 | G | A | 3651 | 0.144 | *atpA* | LSC |
| 22052 | C | T | 3783 | 0.144 | *rpoB* | LSC |
| 9142 | C | A | 3700 | 0.140 | *atpA* | LSC |
| 9137 | T | G | 3701 | 0.139 | *atpA* | LSC |
| 32024 | G | A | 3374 | 0.138 | *psbC* | LSC |
| 9119 | T | C | 3617 | 0.137 | *atpA* | LSC |
| 22169 | C | T | 3841 | 0.136 | *rpoB* | LSC |

*Gentiana* showed that *G. tongolensis* has a similar genome structure and gene order with *G. straminea* and *G. stipitata* (*Fu et al., 2016*; *Ni et al., 2016*; *Sun et al., 2018*), but had partial gene loss in *petD* (Fig. S1).

## Character of SNPs

After filtering with variant frequency and Q-score, a total of 112 SNPs and 31 indels were identified in the plastome of *G. tongolensis*. The average sequencing depth of each base at a variant position was 2699x. Among the variant positions, 45 (31.46%) had a frequency more than 10%, and all were located in the LSC region. A total of 22 SNPs or indels were detected in the SSC region, and none in the IR regions. The top 20 SNPs or indels with highest frequencies are presented in Table 2. After locating the 143 sites and function annotation, 96 were located in genes, including three in introns, and 47 were located in the intergenic space (Fig. 1; Table S2). Forty-four out of 45 positions whose SNP frequencies were more than 10% were located in genes *atpA*, *psbB*, *rpoB*, *rpoC1*, *trnH*, *psbC*, *rpoA* and *psaA*. There were 22 C→T plus 15 G→A transitions, and 16 T→C plus 23 A→G transitions in the plastome of *G. tongolensis*, and thus no excess of mutations was attributable to $5^{m}C$ deamination which is a clue as to whether the sequence is nucleus-derived or plastid-derived (*Huang et al., 2005*).

**Table 3 Summary of verified Sanger sequenced data with Illumina sequenced data in a single individual of *Gentiana tongolensis*.**

| Position in plastome | 23,411 | 68,770 |
|---|---|---|
| Function | *rpoB* | *psbB* |
| Illumina sequencing data | C/T | ATTGAATT/———————— |
|    Frequency of SNP/indel | 0.145 | 0.153 |
| Sanger sequencing data | C/T | ATTGAATT/———————— |
|    Frequency of SNP/indel | 0.043 | 0.038 |

## Verification and genetic characteristics of SNPs

To verify the polymorphisms detected in the Illumina data, we selected three regions that spanned multiple polymorphisms and two control regions without polymorphisms for cloning and Sanger sequencing (Fig. 1). Verification was first performed for the one individual for Illumina sequencing. In the two regions used as positive controls, 10 and 12 clones were sequenced successfully in GTcpSNP-1 and GTcpSNP-2, respectively, and did not show any SNPs. In GTcpSNP-3, a total of 22 clones were successfully sequenced, and nine SNPs and one indel were detected (Table S3). In GTcpSNP-4, 23 clones were sequenced, and three SNPs were detected (Table S4). In GTcpSNP-5, 26 clones were sequenced, and five indels and two SNPs were detected (Table S5). Focusing on the individual for Illumina sequencing, only two sites (a SNP, position 23411; a indel, position 68770) in the sequences obtained from verification were matched with the sites in the plastome on which our primers were designed (Table 3). On the other hand, lots of new variable sites were detected through verification.

For further clarifying the SNPs in the plastome, four more individuals were verified and analyzed with the one individual for Illumina sequencing. For GTcpSNP-1 and GTcpSNP-2, 10 and seven clones were sequenced in two individuals. For GTcpSNP-1 and GTcpSNP-2, a total of 20 sequences with 636 bp and 19 sequences with 571 bp were obtained from three individuals, respectively. No SNP was detected in the two positive control regions in all verified individuals. In GTcpSNP-3 and GTcpSNP-4, a total of 25 and 29 clones were sequenced in two and four individuals, respectively. In GTcpSNP-3, there were 47 obtained sequences ranging from 775 to 778 bp in length. Nucleotide variations were detected at 39 sites, with 35 base substitutions and four indels, and a total of 22 haplotypes were identified in three individuals (Table S3). In GTcpSNP-4, 52 obtained sequences were 614–615 bp in length. Nucleotide variations were detected at 21 sites, with 19 base substitutions and two indels, and 15 haplotypes were identified in five individuals (Table S4). In GTcpSNP-5, there were 26 obtained sequences ranging from 562 to 572 bp in length. Nucleotide variations were detected at seven sites which included five indels and two SNPs, and a total of eight haplotypes were identified in one individual (Table S5). Besides the same sequences with the reference plastome were shared by different individuals, no haplotype was shared among individuals, indicating high heterozygosity in the plastome. We attempted to determine the phylogenetic relationship

among haplotypes, but we found no phylogenetic structure among them (results not shown).

## DISCUSSION

Polymorphisms are common in plastid genomes at the level of population or species, but intra-individual heteroplasmy has rarely been reported (*Medicago sativa*, *Fitter et al., 1996*; *Cynomorium coccineum*, *García, Nicholson & Nickrent, 2004*; *Senecio vulgaris*, *Frey, Frey & Forcioli, 2005*; *Silene vulgaris*, *McCauley et al., 2007*). Due to the modular nature of plant growth, perennial plants can develop the same type of organs independently and at different times during their lifetime, thus leading the intra-individual genomic variation, for example in apples (*Gillian et al., 2017*); however, this does not explain the heteroplasmy in *G. tongolensis*, which is an annual herb. Another possibility of the heteroplasmy is due to DNA damage—either DNA damage due to technical reasons (e.g., sample storage) or to biological reasons. DNA damage during sample storage with silica gel maybe one potential artificial factor leading to the appearance of SNPs in Illumina read data. However, the agarose electrophoresis before library construction did not show any signs of degradation. Besides, DNA damage that occurred due to biological processes would lead to random mutations whose frequencies should be very low. However, our data showed that SNPs were mainly located in genes in the LSC region, and the frequencies of a number of SNPs were high. On the other hand, compromise in plastid DNA repair is a potential reason that cannot be excluded by our present data. Because the rate with which we have detected variants among clones within a verified individual was well above expected somatic mutation levels, it seems unlikely that this is a cause for the heteroplasmy in *G. tongolensis*.

The transfer of plastid DNA, ranging from individual genes (e.g., *rpl22*, *Jansen et al., 2010*; *rpl32*, *Park, Jansen & Park, 2015*) to as much as 131 kb in rice (*Noutsos, Richly & Leister, 2005*), is an ongoing and ubiquitous process in plants (*Martin et al., 1998*; *Huang et al., 2005*). Plastid DNA insertions in the nucleus are in a new mutational environment and have different evolutionary fates compared with plastid genomes (*Huang et al., 2005*; *Noutsos, Richly & Leister, 2005*; *Michalovova, Vyskot & Kejnovsky, 2013*). Chloroplast genomes exhibit very low levels of cytosine methylation (*Ayliffe, Scott & Timmis, 1998*), which is extensive in plant nuclear genomes (*Finnegan et al., 1998*). Therefore, an excess of mutations attributable to $5^{m}C$ deamination, detected by $C \rightarrow T$ plus $G \rightarrow A$ transitions vs. $T \rightarrow C$ plus $A \rightarrow G$ transitions, is a clue as to whether the sequence is nucleus-derived or plastid-derived (*Huang et al., 2005*). Because no excess of mutations attributable to $5^{m}C$ deamination were observed in this study, it seems that we may not attribute the individual mutations to the transfer of plastid DNA to the nucleus in *G. tongolensis*. However, more evidence is necessary to uncover the reason for the heteroplasmy.

These findings raise fundamental questions about chloroplast inheritance in *Gentiana*. Among the seed plants, chloroplasts can be inherited strictly from the female parent, strictly from the male parent, or biparentally (*Mogensen, 1996*). Most flowering plants exhibit maternal plastid inheritance, but there is rare paternal plastid inheritance among such plants as *Medicago* (*Schumann & Hancock, 1989*) and *Arabidopsis*

(*Azhagiri & Maliga, 2007*). However, some genera display biparental plastid inheritance such as in *Coreopsis grandiflora* (*Mason, Holsinger & Jansen, 1994*), *Chamaecyparis obtusa* (*Shiraishi et al., 2001*), *Medicago truncatula* (*Matsushima et al., 2008*), and Caprifoliaceae (*Hu et al., 2008*). Paternal, maternal, and biparental plastid inheritance have even been detected from both interspecific and intraspecific crosses in *Passiflora* (*Hansen et al., 2007*). Biparental plastid inheritance is a reasonably widespread trait (20% to nearly one-third) in angiosperms (*Smith, 1989*; *Zhang & Sodmergen, 2010*), and it is probable that heteroplasmy occurs on a limited scale in most groups of angiosperms (*Azhagiri & Maliga, 2007*; *Hansen et al., 2007*). Therefore, this suggests that intra-individual heterozygosity in the *G. tongolensis* plastome may be observed in functionally paralogous copies within an individual caused by biparental plastid inheritance. Consistent with this idea is the fact that all high-frequency SNPs detected in the plastome of *G. tongolensis* were located in coding regions rather than intergenic regions.

Although the plastome is conserved in flowering plants, structural variations such as gene loss, IRs, and rearrangements are common (*Downie & Jansen, 2015*; *Ruhlman & Jansen, 2018*). Recent studies in *Gentiana* show that the plastome structure is conserved, except for the *ndh* gene being lost in section *Kudoa* (*Fu et al., 2016*; *Sun et al., 2018*). In the plastome of *G. tongolensis*, we detected partial loss of *petD*, which encodes subunit IV of the cytochrome $b_6/f$ complex (*Esposito et al., 2001*). We also found that the *psbB*, *trnH* and *rpoB* genes have distinctly high frequencies of SNPs within an individual in this study. Intra-genomic synonymous rate variations resulting from localized variations in mutation rate can lead to the distribution of SNPs among the coding region (*Zhu et al., 2014*). However, *psbB*, *trnH* and *rpoB* were not mutational hotspots in *Gentiana* (Fig. S1; *Sun et al., 2018*). Although locus-specific substitution rates are not based on gene or protein function (*Zhu et al., 2014*), the possibility that the three genes may have experienced or are experiencing unusual evolutionary processes cannot be excluded. Serious mutations in the plastid genome may cause nuclear-plastid incompatibility, which is the most important driving force for altering modes of organelle inheritance (*Hurst, Atlan & Bengtsson, 1996*). One hypothesis suggested is that biparental plastid inheritance may have been derived to rescue angiosperms with nuclear-plastid incompatibility caused by defective plastids (*Zhang & Sodmergen, 2010*). The case of *G. tongolensis* may support this hypothesis, but more evidence is required.

Plastid DNA is a ubiquitous molecular marker in plant systematics and population genetics research, such as DNA barcoding (*CBOL Plant Working Group, 2009*) and plant phylogeography (*Morris & Shaw, 2018*), because of its uniparental inheritance and haploid nature in flowering plants (*Wolfe, Li & Sharp, 1987*; *Shaw et al., 2014*). In most angiosperms, plastids are under maternal inheritance, thus plastid datasets can provide information on past changes in species distribution when the colonization of new habitats occurs through seeds (*Petit et al., 2003*). They also offer insight into plant phylogenetic relationships by eliminating complex evolutionary events such as hybridization or introgression. Plastomes have also been widely applied for reassessing classifications, divergence dating, and mutational hotspot identification (*Nikiforova et al., 2013*; *Yao et al., 2019*). Biparental inheritance of plastids will result in divergent plastids within an

individual and are functionally paralogous (*Wolfe & Randle, 2004*), thus random sampling of nucleotide sequences from different plastid haplotypes or from paralogous sequences may affect the resulting gene tree topology and the resulting inferences (*Wolfe & Randle, 2004*; *Hansen et al., 2007*). Examples have been presented in the phylogenetic inferences in *Passiflora* (*Hansen et al., 2007*) and *Picea* (*Sullivan et al., 2017*). Since we detected intra-individual plastid genome heterozygosity in *G. tongolensis*, the chloroplast phylogeny may be problematic for phylogenetic inference and not accurately represent ancestor–descendant relationships, especially in *Gentiana*. Besides phylogenetic studies, intra-individual chloroplast heterozygosity can also affect population studies that are sometimes based on the inference of genetic diversity within and between populations. Therefore, before chloroplast heteroplasmy is further studied in more taxa, additional attention should be paid when chloroplast sequences are used.

## CONCLUSIONS

In summary, we reported a case of heteroplasmy and its characteristics in the *G. tongolensis* plastome, but the underlying cause of it is unclear from the present data. Biparental plastid inheritance and transfer of plastid DNA seemed to be most likely, and to better address this question, it will be necessary to collect more data from this species. Hopefully, this study will encourage botanists to reconsider the heredity and evolution of chloroplasts and exercise caution when using them as genetic markers, especially in *Gentiana*.

## ACKNOWLEDGEMENTS

We thank the three anonymous reviewers and the editor for their helpful comments and suggestions on the manuscript.

### Funding
This work was supported by the National Natural Science Foundation of China (Nos. 31600296, 31870697). The funders had no role in study design, data collection and analysis, decision to publish, or preparation of the manuscript.

### Grant Disclosures
The following grant information was disclosed by the authors:
National Natural Science Foundation of China: 31600296, 31870697.

### Competing Interests
Zhi-Cheng Long is employed by HostGene. Co. Ltd.

### Author Contributions
- Shan-Shan Sun conceived and designed the experiments, performed the experiments, analyzed the data, contributed reagents/materials/analysis tools, prepared figures and/or tables, approved the final draft.

- Xiao-Jun Zhou performed the experiments, authored or reviewed drafts of the paper, approved the final draft.
- Zhi-Zhong Li analyzed the data, contributed reagents/materials/analysis tools, authored or reviewed drafts of the paper, approved the final draft.
- Hong-Yang Song performed the experiments, prepared figures and/or tables, approved the final draft.
- Zhi-Cheng Long analyzed the data, prepared figures and/or tables, approved the final draft.
- Peng-Cheng Fu conceived and designed the experiments, performed the experiments, prepared figures and/or tables, authored or reviewed drafts of the paper, approved the final draft.

## DNA Deposition

The following information was supplied regarding the deposition of DNA sequences:

Raw data from Illumina are available at the Genome Sequence Archive of the BIG Data Center: CRA001588.

The plastome of *Gentiana tongolensis* are available at GenBank: MK251985.

The sequences of verified loci are available at GenBank: MK251953–MK251984, MN602314–MN602324.

## Data Availability

Data is available at Figshare: Fu, Peng-cheng (2019): Sequences of three regions identified in *Gentiana tongolensis*. figshare. Dataset.
DOI 10.6084/m9.figshare.7723247.v1.

## Supplemental Information

Supplemental information for this article can be found online at http://dx.doi.org/10.7717/peerj.8025#supplemental-information.

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
