# Peer review of "Intra-individual heteroplasmy in the Gentiana tongolensis plastid genome (Gentianaceae)"

_PeerJ, doi:10.7717/peerj.8025_

## Round 0.1 · original submission · Major Revisions

· Academic Editor

Major Revisions

After careful review of the manuscript and the reviewer reports I believe there are several areas that could benefit from additional clarification.

1) Overall the manuscript is well-written but some typos and grammatical errors are present;
2) the methods section will need additional information regarding assembly parameters etc. (please pay particular attention to Reviewer's 2 and 3), why the validation sequence primers failed, how were the positive clones sequenced (e.g. Sanger);
3) consistency and clarification of some aspects of terminology (please pay particular attention to Reviewer's 2 and 3);
4) acknowledging that there may be other artifacts contributing to the signal of variable SNPs within the plastid genome - the artifacts should be discussed as a possibility at least, but preferably some arguments as to why they are unlikely would also be offered.

Reviewer 1 ·

Basic reporting

Professional English was used in the manuscript. sufficient background of the sytudy was also well desceibed. Professional article structure, figures and tables were aslo well provided. All data were submited to GenBank. And results have been discussed and analyzed.

Experimental design

The experiment of manuscript is well designed based on a interst research question. Methods described with sufficient detail & information to replicate.

Validity of the findings

The manuscript reported intra-individual plastid genome heterozygosity in Gentiana tongolensis, which will offer a new view for botanists to reconsider the heredity and evolution of chloroplast and be cautious with using chloroplast as genetic markers, especially in Gentiana.

Additional comments

The manuscript reported intra-individual plastid genome heterozygosity in Gentiana tongolensis, which will offer a new view for botanists to reconsider the heredity and evolution of chloroplast and be cautious with using chloroplast as genetic markers, especially in Gentiana. As the first report about palstome heterozygosity in plants, it meet the aims and Scope of the journal, which should be published. Nonetheless, authors need to make some revisions to inprove the manuscript. I marked in the PDF file.

Annotated reviews are not available for download in order to protect the identity of reviewers who chose to remain anonymous.

Reviewer 2 ·

Basic reporting

Sun and colleagues examine the chloroplast of the angiosperm Gentiana tongolensis, they find evidence that individuals in the species exhibit chloroplast heteroplasmy. The authors should be commended for the work and thought they put in to this study.

Overall comments:

I had difficulty following a number of different aspects of the study due to grammatical issues and potentially when certain terms were used. Having a thorough read through for grammar would greatly help improve the manuscript.

The paper seems to focus on this being the first report of plastome intra-individual heterozygosity, but throughout the paper it is mentioned that this has been reported before. If it’s the first report, then that’s great it would be a new phenomenon found in plants. If this is not the first report and confirmatory findings, that’s great too. This study builds off of previous work on the subject and even if it is not the first to break new ground it does not mean the finding is not worth publishing.

Abstract
I believe the first sentence is a bit strong on this “Chloroplast is inherited strictly from the female parent, exhibiting haploid nature in most angiosperms”, there are limited examples of paternal inheritance. Saying something like “often” or “typically” rather than “strictly” would be better.
Example of non maternal:
McCauley, D. E., A. K. Sundby, M. F. Bailey MF, M E. Welch. 2007. Inheritance of chloroplast DNA is not strictly maternal in Silene vulgaris (Caryophyllaceae): evidence from experimental crosses and natural populations. American Journal of Botany 94: 1333–1337.
I believe intra-individual heterozygosity, or heteroplasmy has been documented a number of times so this is not the first. “This is the first report about intra- individual plastid genome heterozygosity in plants, and the cause is unclear from present data, although biparental plastid inheritance seems to be most likely.”
Example of within individual heteroplasmy:

Hensen, A.K., L. K. Escobar, L. E. Gilbert, and R. K. Jansen. 2007. Paternal, maternal, and bipaternal inheritance of the chloroplast genome in Passiflora (Passifloraceae): implications for phylogenetic studies. American Journal of Botany 94: 42–46.

Introduction

I don’t think a citation is necessary in the first sentence and if a citation is used it would preferably be the first to describe the value of the chloroplast.

At line 48 you get into the crux of the study and make a less definitive statement than in the abstract, this would be a good place to put some citations regarding the rare instances of intra-individual heterozygosity.

At line 73 the claim of this being the first report is made again but I don’t think that’s true. The authors are likely better read on the subject but it seems as though other studies have reported this. The authors mention structural heteroplasmy at line 47 and I’m not sure how you could get structural heteroplasmy without having an ancestral population that exhibited intra-individual heterozygosity.

Results

Line 143-144: I’m not really following why it would not be hard? Or why it was hard?

At line 178 is there a reason these could not be supplementary material?

Discussion

I’m a bit confused as to lines 181-182. These state that “intra-individual heterozygosity has been reported” but previously the authors have stated this is the first study to report it.
Line 193: It is stated that biparental inheritance is rare (that seems to contradict previous statements).
Line 200: It is stated that this is a relatively widespread trait (This also seems to contradict previous statements)

Experimental design

Line 96: I could be missing something but I do not see the results of OGdraw or a plastome map?

Line 126: What does it mean by aligned and edited? Which aligner from Genious was used? How were these edited?

Line 128: Which model of evolution was selected as the best?

Validity of the findings

no comment

Additional comments

Hopefully these comments are helpful and can be of assistance with your manuscript. Good luck as it continues forward!

Reviewer 3 ·

Basic reporting

This article does not meet several of the criteria established by PeerJ for basic reporting. Below, I identify the areas that need attention, along with suggested improvements.

1. The language used throughout the manuscript is not clear, unambiguous and professional English. There are also many clear typos. Here are a few examples: “duo” instead of “due” on line 58; “palstomes” instead of “plastomes” on line 68; “palstome” instead of “plastome” again on line 73. Furthermore, there should be more care given to the usage of proper scientific terms. For example “heteroplasmy” is often incorrectly used to describe genetic variation among species or populations (lines 47, 180). Its definition is genetic variation among organellar genomes within a cell or within an individual. The authors apply this term correctly when characterizing their intra-individual genetic variation in plastomes, but not when describing variation within populations or species. In lines 154 – 155, it would be better to use the terms biallelic, triallelic, and quadriallelic sites. In lines 160-161, they describe these as the 1st, 2nd and 3rd codon positions. To solve these problems, I recommend that the authors have their manuscript edited by a scientist who is a native English speaker or a professional scientific editing service.

2. The authors have cited the appropriate literature in most cases, but are still missing a few relevant citations. For example, the sentence in lines 46-48 is missing a citation. The authors cite the literature concerning the transfer of individual chloroplast genes to the nucleus in various plant species in lines 186 - 188, but ignore the vast body of work showing that integrations of the plastid genome into the nucleus can be quite large and contain multiple genes and intergenic sequences (Michalovova et al. (2013) Heredity; Huang et al. (2005) Plant Physiology, and Noutsos (2005) Genome Research). This work is especially relevant to the major finding in this manuscript (which I will discuss in a later section of this review). Finally, in line 135, they say that their plastid genome assembly “constituted a closed circular molecule”. A circular-mapping assembly does not indicate anything about the linearity or circularity of the molecules themselves, as circularly permuted linear molecules or head-to-tail linear concatemers can also produce a circular map and genome-sized circular molecules are rarely, if ever, observed (see Bendich, 2004, Plant Cell). This is actually a pretty common misconception in the field of organelle DNA research and I encourage the authors to note this as a way of correcting this misconception.

3. The structure of the article is professional and appropriate. The authors share the Genbank references for their plastome assembly and various PCR clones, but they do not indicate that they have deposited the raw Illumina read data in the short read archive or any public repository.

4. The article is self-contained, but there are some inconsistencies in the reporting. In the Abstract and Introduction (Lines 32-33 and 73), the authors claim that their study is the first report of heteroplasmy (intra-individual plastome variation). However, they later say in the Discussion that “rare intra-individual heterozygosity has been reported (Medicago sativa, Fitter et al., 1996; Cynomorium coccineum, Garcia, Nicholson & Nickrent, 2004) (lines 181-183). Those statements are directly opposed.

Experimental design

1. This article is an original research article and it is within the Aims and Scope of the journal.

2. The research question is reasonably well-identified and the authors clearly identify their research objective for this manuscript as part of an ongoing larger research project aimed at characterizing the plastomes of the genus Gentiana.

3. This investigation suffers from some technical flaws that may affect the major conclusions they draw from their results. I will detail these insufficiencies in the “Validity of the findings” section.

4. There are a few places in the Methods where more information is need to allow for replication of their study. In lines 89-90, the authors need to include additional BLASTN parameters, such as the word size, gaps allowed, reward for a match and penalty for a mismatch. The assembly parameters for NOVOPlasty should be included in lines 90-91. The methodology for calling SNPs in lines 103 to 105 needs to be more detailed. What criteria were used to determine a SNP? How did they distinguish the quality of the SNP calling? In line 146, how did the authors decide whether a base was low quality or not? In lines 147-148, I am not sure what is meant by “among the 145,757 nucleotides of G. tongolensis plastome, SNPs were counted in 100,260 nucleotides (68.78%).” I think it would be better to just mention how many of the total 145,757 nucleotide positions had a SNP.

Validity of the findings

1. The authors conclude that the existence of hundreds of SNPs at >1% frequency in the plastome of Gentiana tongolensis is an indication of heteroplasmy. In my opinion, the authors have not effectively ruled out alternative possibilities that explain their data. Thus, the data presented in the manuscript are not robust, statistically sound and controlled. Here, I will outline several alternative explanations and make suggestions for how the authors can rule these out.

Alternative explanation #1: Sequencing errors / errors in SNP calling
While the error rate of Illumina sequencing is often low, it is highly variable and can be affected by the Illumina platform used, whether a base is in read 1 or read 2, and whether the base is near the end of the read. In some situations, the error rate can approach 1-2% (see Schirmer et al. 2016 Bioinformatics). From Table S2, many of the SNPs reported as >1% are indeed very close to the error rates described for the ends of reads and could be sequencing errors. Also, false-positive SNPs can occur frequently due to misalignments of the reads or having a position only covered by the ends of reads. The authors attempt to confirm these SNPs through Sanger sequencing of four “loci”. It is not clear how they are defining a locus in these cases. It sounds like each of these “loci” may span several SNPs, but in other places in the manuscript, they use the term “loci” to refer to individual SNP positions. From context, I gather that these four sequenced “loci” span several SNPs. They give the primer sequences used to amplify these “loci”, but it is hard as a reader to match their “loci” with the SNP positions in their plastome assembly (i.e. those listed in Table 2 and Table S2). The authors state that “four loci with a high frequency of SNPs were randomly chosen” on line 162. It's not clear if they mean that each locus spans several SNPs and they are all high frequency (i.e. more than 10%) or if only some of them are. It would be better to include a table giving the starting and ending coordinates of each of the four “loci” and list the positions and frequencies of SNPs that fall within the region amplified by their primers. They later describe genes that are included in the GTcpSNP3 and GTcpSNP4 loci, but they do not provide any information about what SNPs were found in the Illumina data and which ones they were able to confirm by Sanger sequencing. The authors then clone the PCR products amplified for each of these four “loci” and do Sanger sequencing of variable numbers of clones for each. For two of them (GTcpSNP-1 and GTcpSNP-2), they did not find any SNPs among the clones that they sequenced. The numbers of clones sequenced in both cases were quite low, so it is possible that the SNPs (even if >10% per individual) would not be sampled among the clones. Or it could be that these are SNPs that failed to be verified by cloning and sequencing. It's not clear. They were able to recover SNPs in the third locus, GTcpSNP3, from 25 sequenced clones among 2 individuals. Was one of these individuals the same one that was sequenced to obtain the Illumina data? They report finding 24 SNPs and 7 indels that make up a total of 19 haplotypes. It isn't clear here whether the SNPs they found are the same SNPs that were identified in the Illumina sequencing data or if one can even compare these data to the Illumina data because they might come from different individuals. The authors state in lines 169-170 that the region includes “parts of rps19, whole trnH and parts of psbA”, but there are no SNPs indicated in the Illumina data for these genes in table S3. They also did not report finding any indels in the Illumina sequencing data. Table S3 shows the 19 haplotypes, but they are not aligned to the consensus plastome assembly, making it additionally difficult to interpret whether these data confirm or contradict the Illumina data. For the fourth locus, GTcpSNP4, the authors obtained 35 clone sequences from 5 individuals. In this case, they found 17 SNPs and 2 indels that made up a total of 13 haplotypes for a region that spanned part of the rpoB gene. There were many SNPs at a frequency of >10% that were identified in rpoB in the Illumina data. In this case, they had sequences for 5 individuals, so one of them must be the same individual that gave rise to the Illumina data. However, it is also not indicated anywhere which SNPs were present in the Illumina data and confirmed in their clone sequencing. Further complicating all of this is the fact that cloning also introduces errors. From my own personal experience cloning genes to make transgenic constructs, I find errors in my clones at frequencies that are similar to the frequencies of SNPs and indels reported in the cloning data that are presented by these authors. To really confirm heteroplasmy, the authors should make a clear comparison between their Illumina data and the clones (if indeed the same individual is represented in the Illumina and cloning data). SNPs that are identified in both the Illumina data and in the clones are likely to be real, as getting the same error in both cases is pretty unlikely. From there, the authors can compare different individuals. I suggest not presenting the data in a pie chart, as there are probably only 5 or 6 sequences in total. I suggest presenting an alignment, like those shown in Tables S3 and S4, where the sequences are aligned to the consensus plastome. Indicate the positions that were found to be variable in the Illumina data with a bold letter and then color code the aligned sequences by individual (and clearly indicate which individual was the one that was sequenced).

Alternative explanation #2: Plastid sequences in the nucleus.
The authors do attempt to rule out this possibility in the Discussion, but the explanations they give do not eliminate nuclear-located plastid sequences as the source of the SNPs in this study. The authors acknowledge that transfer of plastid sequences to the nucleus does occur, but suggest that it only occurs at the level of individual genes (lines 186-188). The authors seem to be unaware that many plant genomes contain segments of plastid DNA that are tens of kb in length. They say that they don't think that their SNP data can be explained by plastid sequences in the nucleus because there are SNPs in intergenic regions. However, many plastid-derived insertions in the nuclear genome come from genic and intergenic regions. I direct the authors to read Michalovova et al. (2013) Heredity; Huang et al. (2005) Plant Physiology, and Noutsos (2005) Genome Research to start with. Nuclear plastid DNA sequences (NUPTs) often show a bias toward C-to-T and G-to-A transitions (due to deamination of methylated C nucleotides in the nuclear genome). When I was just casually browsing the SNPs reported in this manuscript, it seemed to me like there were a lot of C-to-T and G-to-A transitions. The best way to rule out NUPTs as a source of the SNPs they observe in their data would be to isolate chloroplasts first and then obtain DNA from the isolated chloroplasts for Sanger sequencing of SNPs to compare with the Illumina data. However, I don't believe that this option is available to these authors because these plants were sampled in the field. What the authors could do is calculate the frequency of C-to-T and G-to-A transitions and determine whether they are indeed the predominant substitutions. This would be a sign that the mutations that occurred due to 5-methyl-cytosine deamination, which would only happen if the sequence was in the nucleus. They could compare this to the pattern of substitutions among the plastid genomes of the other species of Gentiana, which should show a lower frequency of C-to-T and G-to-A transitions. Another experiment would be to do a restriction digest of the total genomic DNA from G. tongolensis using a methylation-sensitive enzyme. They could then do a southern blot and probe the blot with a plastid-derived sequence. If there is a lot of plastid DNA in the nucleus, this would be evidenced by a strong hybridization signal in the uncut fraction of the DNA. It is possible that there are just far more NUPTs in G. tongolensis than in the other Gentiana species studied by these authors, which is why they have not encountered this situation before.

Alternative explanation #3: DNA damage
Is it possible that the DNA is damaged and that this is the reason why so many SNPs appear in the Illumina read data. This could be for technical reasons, such as the silica drying process used to preserve the samples collected in the field or for biological reasons, such as the age of the tissue collected or that perhaps this particular species of Gentiana is compromised in plastid DNA repair. Silica gel has been used reliably with other species, but I have also heard reports of silica gel leading to DNA damage. I would be curious to know whether the authors examined their DNA on a gel before sequencing to verify that it did not show any signs of degradation before sequencing. DNA damage that occurred due to biological processes would still lead to heteroplasmy, but not of the type that the authors propose. In this case, there would be random haplotypes present in an individual, but they are not maintained as distinct haplotypes. It is hard to distinguish between these possibilities given the data they present.

Additional comments

This study suggests the possibility of heteroplasmy that is maintained within individuals of a species, which is really interesting. However, there are several alternative explanations that cannot be ruled out given the current state of the data and the analysis. I would really like to see this paper be eventually published and I encourage the authors to follow through on the suggestions I have outlined in my review.

---

## Round 0.2 · Major Revisions

· Academic Editor

Major Revisions

One of the reviewers has indicated that there is still a major technical/methodological issue with the manuscript. The issue lies with the ability to determine if the variants are real or artifacts of sequencing technologies. The reviewer has, very kindly, provided some suggestions for how to move forward with clarifying this issue. Please pay particular attention to these comments in your revisions.

In addition, a number of spelling and grammatical errors remain. Please try to correct these.

Reviewer 3 ·

Basic reporting

The authors have greatly improved the quality of the language and have adopted the correct usage of scientific terms in the revised article. However, some typos still remain and there are some areas where more clarification is needed. These are:
Line 59: I am not sure what is meant by "(title = chloroplast AND complete genome). Are these search terms used to identify how many complete chloroplast genome sequences there are in Genbank?
Line 111: "111 Reads were BLASTED with the assembled plastome using BWA-MEM (Li, 2013) ...". Usually when people use "BLAST" as a verb, they are referring to using BLAST to align sequences. From the second part of the sentence, it looks like the authors used BWA to align the sequences. If BWA was used, then "BLASTED with" should be changed to "aligned to".
Line 192: "one individuals" should be changed to "one individual".

The cited references and context provided are sufficient. The structure of the article is professional. The authors indicate that the raw data have been shared. However, I cannot retrieve the assembly from Genbank using the accession number MK251985. The authors deposited the raw read data in the National Genomics Data Center, BIG Data Center under accession number CRA001588 in response to my last request and I was able to easily locate the files.

The article is self-contained, but the relationship between the results and the hypotheses is still unclear. Please see more details about this in the "Validity of the findings" section.

Experimental design

No comment.

Validity of the findings

I appreciate the authors' efforts to respond to my earlier comments. This revision has improved a lot, but I have still identified one major issue that needs to be resolved before publication in PeerJ.

Regarding the statements in lines 192-196:
"Focusing on the individual for Illumina sequencing, only two sites (a SNP, position 23411; a indel, position 68770) in the sequences obtained from verification were matched with the sites in the plastome on which our primers were designed. On the other hand, lots of new variable sites were detected through verification."

This statement is a little unclear, but I have been through the data and I think I know what they mean. The authors selected three regions, each of which covered more than one SNP or indel that had been detected in the Illumina data. Only two (of many) sites (the SNP and the indel they mentioned in the statement) were also identified in the Sanger data. In addition, they found many more sequence variations in the Sanger clones that were not observed in the Illumina data.

Since the same individual was used both for the Illumina sequence and the Sanger sequencing, I would expect a complete overlap between the two datasets. However, there is variation present in the Illumina data that is not present in the Sanger data and there is variation in the Sanger data that is not present in the Illumina data. If the disagreements between the two datasets was only a minor fraction of the variable sites, then I think that this would indicate that there is true biological variation with some variation due to technical artifacts. Because there is so little overlap between the two datasets, it is really hard to tell what this means.

One explanation for the failure to detect polymorphisms that had been observed in the Illumina data in the Sanger sequences is low number of clones per individual. For example, in GTcpSNP-4, there should have been two SNPs that had been detected in the Illumina data present in the Sanger data. I am inferring this by cross-referencing the coordinates given in the Materials and Methods with Table S2. The authors should make it easier to find this information, perhaps by highlighting the SNPs in Table S2 with color-coding that indicates which SNPs are covered by which GTcpSNP regions that were Sanger sequenced. Those two SNPs should have an allele frequency of around 15%, meaning that each should be present in 3 of 20 clones (15%) for a single individual. The authors only sequenced two clones for the individual that was used for the Illumina sequencing, so it is not surprising that only one of these two SNPs was detected. They were lucky to find one of the SNPs among only two clones, given that the probability of this happening is low.

Similarly, I inferred that GTcpSNP-5 should have contained 5 SNPs and 2 indels. According to Table S2, the allele frequencies for these polymorphisms ranged from 8.9% to almost 26%. Only 6 clones were sequenced for this individual, meaning that at best only the polymorphism with the highest frequency (an indel at position 68516) would be found in one or two of the six clones. However, it was the indel at position 68770 (listed at 68769 in TableS2) that had a lower allele frequency that was detected in the Sanger data.

I was unable to make a similar comparison for GTcpSNP-3 because the coordinates were not given in the Materials and Methods (although the coordinates were given for all of the other regions). Cross-referencing the positions in Table 3 with Table S2 did not help because none of the positions in Table 3 were also found in Table S2. Also, the genes in the GTcpSNP-3 region (rps19, trnH, and psbA) are not listed in Table S2. This needs to be clarified.

It is difficult to know how to interpret the polymorphisms that were present in the Sanger data that were not found in the Illumina data. I think that the most likely explanation is that these are mutations that were introduced during the cloning process. If that is the case, then why are there so many? I would expect maybe a few here or there, but I would not expect nearly all clones to have an independent artifactual mutation, which is what I see in Tables 3-5. Another possibility is that the Sanger sequencing data is of low quality. In either case, they would be technical artifacts. The only other interpretation is that they represent true biological variation that is somehow missed in the Illumina data, but this seems unlikely given the massively high coverage of the Illumina data.

To summarize, there is very little overlap in the polymorphisms detected in the Illumina data and those detected in the Sanger data. Based on the data presented by the authors, it is not clear which polymorphisms represent true biological polymorphisms and which are technical artifacts.

I would recommend sequencing more clones (at least 20 per individual per region). If the frequency of the mutation (based on Illumina sequencing) is 15%, then there is a probability of 0.85 of not finding it if you sequence only one clone. If you sequence two clones, then the probability of not finding the mutation is (0.85)^2, or 0.72. If you sequence 20 clones, then the probability of not finding the mutation is (0.85)^20, or 0.039. Thus, if the authors sequence 20 clones per individual for each region, then they have a good probability of finding all of the mutations present in the Illumina sequencing data.

If the authors do this and there are still polymorphisms that appear in the Sanger sequencing data from the clones that do not appear in the Illumina data, then I think that the polymorphisms present only in the Sanger data should be dismissed as possible technical artifacts and thus they should not be scored as new haplotypes.

Additional comments

I have a few more suggestions for the authors to improve clarity and the presentation of their data.

1. Please consider presenting Figures S1 and S2 as main figures instead of supplemental figures. There are no figures in the main document and adding these visual representations of the data would be a welcome addition.

2. For the plastome map in Figure S1, it would be great if the authors could also indicate the regions that were cloned and subjected to Sanger sequencing.

3. For the plastome map in Figure S1, the authors should consider plotting the locations of the SNPs.

4. Please consider making a table of the coordinates for each of the 5 regions used in the cloning and Sanger sequencing analysis.

5. Line 164: I don't think it is necessary to have two decimal places for the depth of coverage, as this level of precision is not meaningful. I would change "2698.72" to 2699x (with "x" indicating that it is 2699-fold coverage).

6. Lines 171-172: Thanks for including this analysis as a response to my earlier critique. I think that there needs to be more clarification here. I think it is necessary to mention why an excess of C -to-T and G-to-A transitions in comparison to T-to-C and A-to-G transitions would be an indication that these are polymorphisms that occur due to nuclear integration.

7. Line 173: More explanation is needed here. It could be something along the lines of "To verify the polymorphisms detected in the Illumina data, we selected three regions that spanned multiple polymorphisms and two control regions without polymorphisms for cloning and Sanger sequencing. You could then refer to a table of the coordinates if you do make this table, as I suggested earlier.

8. Lines 176-178: It is not clear what this means. If you detected 7 SNPs among 12 clones, that could mean that there was one clone out of 12 that had 7 SNPs in it. Or it could mean that 7 of the 12 clones each had one SNP. Or it could mean anything in between. The language for all of the results described in these lines needs to be clarified so there is only one possible meaning.

9. Lines 208-212: Some rephrasing is needed here. Thanks for including this discussion in response to my earlier critique. However, I think that it needs to be stated at the beginning that you are evaluating the possibility that the heteroplasmy is due to DNA damage - either DNA damage due to technical reasons (sample storage) or to biological reasons. Then present the reasons why you are ruling out these possibilities.

10. Table S2: Please indicate which are located within the regions used for cloning and Sanger sequencing. Also, please revise the language in the header. "Allelic depths of Reference Base" can simply be changed to "Depth of coverage". Finally, I don't see any entries in the "location" column that say "intergenic" even though you mentioned in the text that there were 45 polymorphisms in intergenic regions.

11. Table 1: the term "Rate (%)" is unclear. Consider changing it to "Percentage of polymorphic sites" or "% polymorphic". Add (bp) to the "Length" in the header.

12. Table 2: Change header to match Table S2.

---

## Round 0.3 · Minor Revisions

· Academic Editor

Minor Revisions

The manuscript has been greatly improved with respect to the reviewer concerns. There are a few small details left to attend to (see below). Also, in the discussion I think you should make it very clear that the rate with which you are detecting variants among clones within an individual is well above expected somatic mutation levels thus, it seems unlikely that this is a cause for the intra-individual variation. At present, the discussion seems to leave the reader to infer that information from the read depths etc. presented in the results - which not all readers might infer. Stating this explicitly will ensure that every reader leaves the paper with that information. Finally, the mention of biparental inheritance raises a question. Are the rates of variation you are observing consistent with (perhaps) what would be expected under biparental inheritance, or are they even higher? I'm not sure of the answer myself, but it seems that they may be higher. Perhaps there is some information that you have come across that might be worth adding to the discussion around this particular point?

Line 52 - data is plural. Please change is to are.
Line 98 - remove with from 'with setting'.
Line 110 - when it says reads were aligned to the assembled plastome, is this referring to the reads used to make the de novo assembly which were then collapsed into a consensus? Or is this now referring to reads from additional specimens of G. tongolensis? It seems that it would be the former (same individual) based on the flow of the manuscript but it would be good to clarify this.
Line 127 - how many other individuals exactly?
Line 155-156 - remove the word 'overall'.
Line 191 - how were 19 sequences obtained if seven clones each from two individuals were run? Shouldn't this be 14? Were some additional clones or replicates sequenced?
Line 329-330 - change 'or is experiencing' to 'or are experiencing'.
Line 342 - change 'due to' to 'by'.

---

## Round 0.4 · accepted · Accept

· Academic Editor

Accept

Thank you for your careful corrections throughout the manuscript. I agree that the biparental inheritance topic is extremely interesting but, unfortunately, lacks data in the literature to make good conclusions from. This would be a really neat area of study for the future though. Thank you for the acknowledgement also. It is my pleasure to help.